# Making a (Counterfactual) Difference
# One Rationale at a Time

**Mitchell Plyler**
Department of Computer Science
North Carolina State University
mlplyler@ncsu.edu

**Michael Green**
Laboratory for Analytic Sciences
magree22@ncsu.edu

**Min Chi**
Department of Computer Science
North Carolina State University
mchi@ncsu.edu

## Abstract

Rationales, *snippets of extracted text* that explain an inference, have emerged as a popular framework for interpretable natural language processing (NLP). Rationale models typically consist of two cooperating modules: *a selector* and *a classifier* with the goal of maximizing the mutual information (MMI) between the "selected" text and the document label. Despite their promises, MMI-based methods often pick up on *spurious* text patterns and result in models with nonsensical behaviors. In this work, we investigate whether *counterfactual data augmentation (CDA)*, without human assistance, can improve the performance of the selector by lowering the mutual information between spurious signals and the document label. Our counterfactuals are produced in an unsupervised fashion using class-dependent generative models. From an information theoretic lens, we derive properties of the unaugmented dataset for which our CDA approach would succeed. The effectiveness of CDA is empirically evaluated by comparing against several baselines including an improved MMI-based rationale schema [19] on two multi-aspect datasets. Our results show that CDA produces rationales that better capture the signal of interest.

## 1 Introduction

Research in neural model interpretability has been cast as important and received significant recent attention [21]. Within the field of natural language processing (NLP), *rationales* have been a popular method for providing interpretability in the form of extracted subsets of text [10]. Rationale models typically consist of two cooperating modules where one module, the "rationale selector", selects the rationale from a source document, and the other module, the "classifier", acts on only the selected rationale without seeing the rest of the document. There is interpretability through sparsity and exclusivity.

A common approach for training these rationale models is based on the maximum mutual information criteria (MMI) [7]. With the MMI criteria, rationale selectors seek the subset of text that carries the most information about the target label. Often, sparsity and coherency constraints are used to keep the rationales interpretable. Within many datasets, however, *spurious patterns* and *co-varying aspects* can cause the rationale selector to pick up on patterns that do not capture a desired relationship between input text and target labels. As a result, the rationalized model can have undesirable behaviour like predicting a hotel is very clean because it is in a convenient location. Nonsensical rationales or

35th Conference on Neural Information Processing Systems (NeurIPS 2021).

explanations might decrease trust in the model, and in some cases, suggest the model might generalize poorly [26].

In this work, we propose a general *counterfactual data augmentation (CDA) [22] approach* to aid rationale models trained with MMI. We show that theoretically our CDA approach can effectively improve the performance of rationale selectors by lowering the mutual information between spurious signals and aspects of interest. Empirically, we show that models trained on our CDA datasets learn higher quality rationales than those trained on the original dataset when both use the same MMI criteria. More importantly, *the most significant advantage of our CDA approach is that it does not require human intervention*. We use rationales from an initial, noisy model and replace them with new text that changes the target label using a generative neural model. In this way, our CDA approach is completely hands off and does not need input from human experts or crowd workers.

We first show that in the extremely ideal scenario where the initial rationale selector is perfect, our CDA approach can eliminate the mutual information between spurious signals and the target label. Next, we show in the common, realistic scenario where the rationale selector is noisy and imperfect, our CDA approach can still yield gains. Finally, the effectiveness of our CDA approach is compared against several baselines including an improved MMI-based rationale schema [19] on two common *multi-aspect* review datasets, *TripAdvisor* [29] and *RateBeer* [24]. Multi-aspect datasets are our main focus as they are guaranteed to contain *spurious patterns* and *co-varying aspects*; we primarily used MMI-based baselines because the fundamental goal our CDA approach is to reduce the mutual information between spurious signals and aspects of interest.

## 2 Counterfactual Data Augmentation and Multi-Aspect Datasets

### 2.1 Definitions and Notations

We use upper-case letters to denote random variables, $X$ and $Y$, and lower-case letters to denote samples from these variables, $x$ and $y$. $I(X, Y)$ marks the mutual information between $X$ and $Y$. Mutual information is defined as the reduction in uncertainty of a random variable due to knowledge in another random variable, $I(X, Y) = H(Y) - H(Y|X)$, where $H(X)$ and $H(Y|X)$ are the Shannon entropy and conditional entropy respectively [8].

### 2.2 Problem Formulation

This work follows the same rationale concept introduced in [19]. Specifically, one neural module extracts text from a document and another neural module classifies the extracted text. Later it has been shown that the rationalization criteria aims "to maximize the mutual information between selected features and the response variable" [7] defined as:

$$\max_G I(X_M; Y) \quad \text{subject to} \quad M \sim G(X) \tag{1}$$

where $M$ denotes a binary rationale mask over the input produced by a rationale selector $G$. That is, the goal of the rationale selector is to select the subset of features in $X$ that are *most informative* of the label $Y$ under some constraints defined by the selector $G$. In this work, our selector constraints affect the size and coherency of the rationales.

Here we consider a multi-aspect dataset, $D$ with features $X$ and labels $Y$: $X$ is a set of features, or a sequence of words in NLP, and $Y$ is a vector, possibly one dimensional, of numerical scores. In a multi-aspect dataset, a single document can discuss multiple attributes of a single object. For example, a single beverage review might discuss its appearance, taste, and smell. We assume some subset of the features, $X_1$, belong to the target aspect label, $Y_1$, while other features, $X_2$, are spurious or non-causal. These features could belong to other aspects or be artifacts of the dataset. In the following, for simplicity, we will use $<X, Y_1>$ or $<X_1, X_2, Y_1>$ interchangeably based on the context.

Our goal is to estimate or model the score for aspect $Y_1$ by using the function $f$ given *only* $X_1$, as follows: $Y_1 = f(X_1)$. There is one such model corresponding to each $< X_1, Y_1 >$ pair. In this work, we use multi-aspect datasets, and model each aspect individually. This simulates the more common case where a dataset has a single output label of interest and all signals in the dataset that do not pertain to that label are considered spurious or belonging to other, not-estimated aspects.

When predicting $Y_1$, our ideal model would focus on $X_1$ and ignore all other features, $X_2$. Specifically, for a given sample <x, y>, our selector would select a subset of $x$ such that $x_1 = x_M$. A rationale selector might fail to extract $x_1$ effectively for two primary reasons: first, Chen et al. [7] showed it is intractable to find a solution to Eqn 1 and thus they derive a variational approximation with Monte Carlo based gradient estimates. Second, datasets can contain artifacts such that spurious patterns might contain significant mutual information with the target label. Along the same lines but even more concerning, other aspects in a dataset might be highly correlated with the target label. For example, beers that smell good usually taste good as well.

Since the goal of the selector is to select features that maximize $I(X_M, Y_1)$, we can assist the selector in finding $X_1$ by lowering the mutual information between the other spurious features and the desired label, $I(X_2, Y_1)$. Drawing on ideas from [22], we do this through a general counterfactual data augmentation (CDA) scheme where, in the counterfactual dataset, superscript $c$, we flip the label of the document from $Y_1$ to $Y_1^c$ and replace the text selected by the rationale selector, $X_1$, with an inference, $X_1^c$, generated by a *class conditioned masked language model (MLM)* using $Y_1^c$ and $X_2$ described as:

$$Y_1^c \leftarrow 1 - Y_1; \quad X_1^c \leftarrow \arg\max_{X_1} p(X_1|1 - Y_1, X_2) \tag{2}$$

In our generated counterfactual dataset $< X_1^c, X_2, Y_1^c >$, $X_1^c$ is the newly generated counterfactual, $X_2$ is the original spurious feature set, and we assign $Y^c = 1 - Y_1$. We will show, in the augmented dataset which is the concatenation of datasets $< X_1, X_2, Y_1 >$ and $< X_1^c, X_2, 1 - Y_1 >$, we are lowering the mutual information $I(X_2, Y_1)$. We will use superscript $a$ for the augmented dataset: $D^a$ = <$X_1^a, X_2^a, Y_1^a$>.

## 2.3 Lowering $I(X_2, Y_1)$, an idyllic case

Take a dataset of beer reviews where each document contains a description of the taste and smell of a beer as well as a numerical score for only the smell aspect. The task is to estimate the smell score while using the smell text as the rationale. Figure 1 demonstrates our process. An example of a concise document in this dataset might be *This beer smells great. It tastes terrific*. In this document $x_1$ is the phrase *smells great* and $x_2$ is *tastes terrific*. Both have positive sentiment but our only label for this document is $y_1 = 1$ for the smell sentiment. Our counterfactual document could be *This beer smells awful. It tastes terrific* and our label becomes $y_1 = 0$. We can see that in the augmented dataset the phrase *tastes terrific*

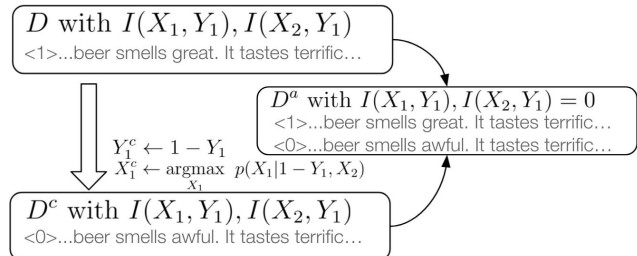

Figure 1: Toy example to demonstrate our approach. In the augmented dataset $D^a$, the mutual information, $I$, between the smell score and the smell text is preserved while the mutual information between the smell score and the taste text is eliminated.

maps to both positive and negative labels and therefore $p(Y_1|X_2) = p(Y_1)$. Finally, $I(X_2, Y_1)$ is 0 and $I(X_1, Y_1)$ is unchanged.

With perfect knowledge of the ground truth rationales and the process $X_1^c \leftarrow p(X_1|1 - Y_1, X_2)$, we can craft counterfactual documents and therefore a counterfactual dataset that perfectly eliminates $I(X_2, Y_1)$ while preserving $I(X_1, Y_1)$. However, the challenge is that ground truth rationales are not provided in the training data. Therefore, it is important for us to show that even when rationales selected by the initial rationale selector are noisy and imperfect, we can still lower $I(X_2, Y_1)$ in the augmented dataset and benefit subsequent models trained with MMI.

## 2.4 Dealing with a Noisy Initial Selector

Here we are working on the augmented dataset: $D^a$ = <$X_1^a$, $X_2^a$, $Y^a$>. To completely eliminate $I(X_2^a, Y_1^a)$, our CDA approach requires a perfect rationale selector. We were originally motivated by improving a poor rationale selector, so here we track what happens to both $I(X_2^a, Y_1^a)$ and $I(X_1^a, Y_1^a)$ when the rationale selector is not perfect. Our new goal is to reduce $I(X_2^a, Y_1^a)$ more than we reduce

$I(X_1^a, Y_1^a)$. We analyze the procedure in the worst case scenario in order to determine a lower bound on our algorithm's benefits under some assumptions.

In an extremely erroneous case, we say that for a given $<x_1, x_2, y_1>$, our initial rationale selector mistakenly selects $x_2$ when aiming for $x_1$. When creating the corresponding counterfactual document, we still have $y_1^c = 1 - y_1$, but we modify the document according to $x_2^c \leftarrow \arg\max_{X_2} p(x_2|1 - y_1, x_1)$ instead of the process defined by Eqn 2. Going back to our concise example, this would be the counterfactual document *This beer tastes good. It also smells bad*. In this extremely erroneous case, we have decreased $I(X_1, Y_1)$ while $I(X_2, Y_1)$ remains unchanged. Thus, we define the worst case scenario as reducing $I(X_1, Y_1)$ at some error rate $\alpha$ and keeping $I(X_2, Y_1)$ constant at the same rate.

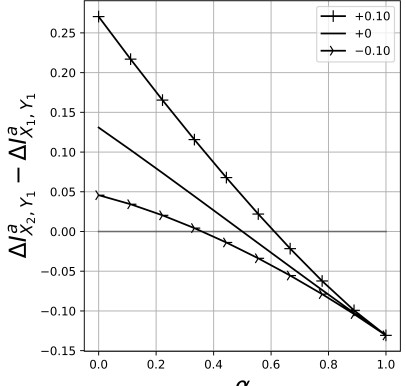

Figure 2: CDA benefits (y-axis) as function of error rate $\alpha$. Each line represents $p(Y_1|X_2) = p(Y_1|X_1) \pm c$.

If we say that this error happens to all samples with a rate $\alpha$, we can analyze conditions that must be present in the original dataset so that our CDA approach is beneficial. Let's first define $\Delta I^a$ as the change in mutual information from the original to the augmented dataset.

$$\Delta I_{X_i, Y_j}^a = I(X_i, Y_j) - I(X_i^a, Y_j^a) \tag{3}$$

In order for our CDA approach to be beneficial, we need to decrease $I(X_2, Y_1)$ more than $I(X_1, Y_1)$. That is:

$$\Delta I_{X_2, Y_1}^a - \Delta I_{X_1, Y_1}^a > 0 \tag{4}$$

For samples in the original dataset where the initial selector results in an error, $X_1$ would map to $1 - Y_1$ for the corresponding counterfactuals. Furthermore, the augmented dataset would map $X_1$ to both $Y_1$ and $1 - Y_1$ for these original erroneous samples and corresponding counterfactuals, and thus, $X_1$ is no longer informative of $Y_1$ on such samples. That is, in the augmented dataset and with proportion $\alpha$, $p(Y_1^a|X_1^a) = p(Y_1^a)$. For the remaining $1 - \alpha$ portion of non-error samples, CDA can successfully capture $p(Y_1|X_1)$ in the original dataset and $p(Y_1^c|X_1^c)$ in the counterfactual dataset. To quantify the conditional distributions of the augmented dataset, we make two assumptions: first, we assume that we can successfully generate the counterfactuals, and thus, we can assume $p(Y_1^c|X_1^c) = p(Y_1|X_1)$; second, we assume that the erroneous samples happen randomly across the original dataset, and the marginal distributions of the original and augmented datasets are the same. Based on these two assumptions, we have:

$$p(Y_1^a|X_1^a) = \alpha p(Y_1) + (1 - \alpha)p(Y_1|X_1) \tag{5}$$

For $X_2$, the success and failure cases are reversed.

$$p(Y_1^a|X_2^a) = (1 - \alpha)p(Y_1) + \alpha p(Y_1|X_2) \tag{6}$$

We can now expand Eqn 4 using the definition $I(X, Y) = H(Y) - H(Y|X)$.

$$0 < -H(Y_1|X_2) + H(Y_1^a|X_2^a) + H(Y_1|X_1) - H(Y_1^a|X_1^a) \tag{7}$$

Using the definition, $H(Y|X) = -E \log p(Y|X)$, we can expand this further to

$$0 < -E \log p(Y_1|X_1) + E \log p(Y_1^a|X_1^a) + E \log p(Y_1|X_2) - E \log p(Y_1^a|X_2^a) \tag{8}$$

Eqn 8 describes the conditions that must be met in our original dataset in order to yield gains from our CDA procedure for some error rate $\alpha$. It is impossible to calculate this relation directly because it will require exact knowledge of our ground-truth rationales. In order to shed some light on this, Figure 2 shows the efficacy of the CDA approach when we approximate $X_1$ and $X_2$ with binary variables, $p(Y_1|X_1) = \frac{3}{4}$, $p(X_1) = p(X_2) = p(Y_1) = \frac{1}{2}$. When $X_1$ and $X_2$ are equally informative of $Y_1$, the benefits of CDA decrease linearly with the error rate, and intuitively, our error rate must be less than $50\%$ to see gains. When $X_2$ is more informative than $X_1$, we have a higher error budget to see any benefit, and when $X_1$ is more informative than $X_2$, our error budget is smaller. According to this analysis, we have a higher error budget when spurious signals offer the same or more information about the target label. When the spurious signals carry much less information, the initial selector must have a low error rate in order to benefit from the CDA approach.

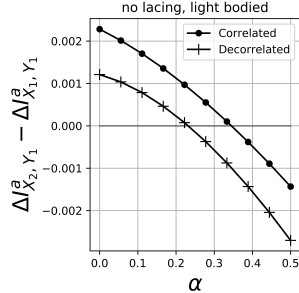

With the datasets used in this work, we can gain insight by approximating $X_1$ and $X_2$ with guessable bigrams. Here is another concise example: $X_1$ is a binary variable that indicates the occurrence of the phrase *no lacing* and $X_2$ is a binary variable for the phrase *light bodied*. For a beer's appearance aspect, *no lacing* is a strong indicator for a negative score, and for its palate aspect, *light bodied* is a strong negative indicator. As beer's appearance is highly correlated with its palatability, both phrases indicate a low appearance score. Taking $Y_1$ as the appearance score, $X_1$ as the occurrence of *no lacing*, and *light bodied* as $X_2$, we can numerically use Eqn 8 to examine how CDA helps us for varying error rates in Figure 3. CDA is beneficial whenever the curve is above zero. In the fully correlated dataset, "Correlated", described in section 4.1, the information carried by *light bodied* is closer to that of *no lacing* than it is in the decorrelated dataset. As a consequence, in the correlated dataset, we have a higher error budget and more opportunities for our CDA.

Figure 3: CDA benefits when approximating $X_1$ and $X_2$ as the occurrence of bi-grams

## 3 Architecture and Implementation

### 3.1 Rationale Framework

The original rationale framework, as viewed in this work, was introduced by [19]. It was not our goal to change the core rationalization algorithm, so we re-implemented the algorithm and updated some details. At a high level, our rationale framework is the same in that we use one network to select the rationale and another network to classify the text. The rationale framework can be visualized by the blue portion of Figure 4.

The original implementation [19] used RNNs for both networks, REINFORCE [30] for dealing with the discontinuity introduced by the binary rationale mask, and variable percentage rationales. In this work, we use transformers for both networks because of their effectiveness over RNNs in NLP [28], the simpler straight-through method [3] [5] instead of REINFORCE, and we use fixed percentage rationales because it eliminates sparsity related hyperparameters. Our fixed percentage rationales differ from Chen et al. [7] in that we use the top-K tokens during training and inference whereas Chen et al. [7] use an iterative re-sampling approach with Gumbel-softmax reparameterization [15].

The classifier is trained only to make quality predictions against the labels while using the rationale. This is the cross-entropy between the labels and the classifier's prediction, $L_y$. We follow [19] by using a coherency regularizer for the rationale selector: $L_r = \frac{\lambda_r}{T} \sum_{1...T} |m_t - m_{t-1}|$ where $T$ is the total number of tokens in a document, $m$ is the rationale mask, and $\lambda_r$ is the hyper-parameter used to tune coherency. This encourages the rationales to be contiguous. For the datasets evaluated in this work, coherency is a useful inductive bias. The loss for the rationale selector $L_s$ is the coherency regularizer and the cross-entropy between the labels and the classifier's prediction. This is $L_s = L_r + L_y$.

### 3.2 Counterfactual Predictor

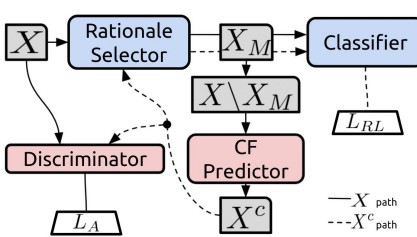

The CDA process described earlier, Eqn 2, requires us to generate new documents with the text and label flipped for just one aspect. That is, we are sampling a new document from $p(X_1|X_2, 1 - Y_1)$. The key challenge is that *we are not provided with ground truth rationales or counterfactuals with which to learn how this data generation process works*. This can be viewed under the lens of unsupervised style transfer for which there is significant prior work in the NLP domain [20] [23]. Our method for generating counterfactual documents leverages many ideas from these works, and our main contribution here is connecting these ideas to the rationale framework. We incorporate the rationale framework in the counterfactual generation process because our goal is to lower the mutual

Figure 4: CF Predictor training flow.

information between the spurious signals and the target label. An off-the-shelf style transfer method might focus on signals other than that selected by our initial rationale selector.

Figure 4 shows the CF Predictor's training flow. At its core, the CF Predictor leverages class-dependent Masked-language models (MLM) [9]. We replace the original document's rationale with an inference from a MLM and leave the rest of the document unchanged. The MLMs are trained to produce documents with the desired class through reinforcement learning (RL) [12], and they are trained to produce realistic documents through adversarial learning [11]. We use straight-through [3] to propagate gradients through token selection during training. For a dataset with binary labels, we train two separate MLMs: one for generating class-0 documents and another for generating class-1 documents [31]. The loss for the counterfactual predictor, $L_{CFP}$ is defined as:

$$L_{CFP} = \lambda_{RL} L_{RL} - \lambda_A L_A \tag{9}$$

where $L_{RL}$ is the classifier loss after passing the counterfactual through the rationale selector and the classifier. It is the cross-entropy between the desired label, all ones or zeros, and the predicted label. $L_A$ is the loss component from adversarial training. Following [11] and [32], our discriminator, $\mathcal{D}$, seeks to distinguish generated documents from the originals, so the discriminator's loss, $L_{\mathcal{D}}$, is $\frac{1}{\lambda_A} L_A$ where $L_A$ is the cross-entropy between real-fake labels and the prediction given by the discriminator. $\lambda_A$ and $\lambda_{RL}$ balance the adversarial and classification losses. We include $\frac{1}{\lambda_A}$ when training the discriminator to hamstring it relative to the predictor. We found that this generally helped us avoid mode collapse commonly seen when adversarially training generative models. For $L_A$, we mask the contribution of original documents without the desired label. When training the class-1 counterfactual predictor, we show the discriminator real documents $X$ with $Y_1 = 1$ and counterfactual documents, $X^c$, where the original documents' labels were $Y_1 = 0$.

During training, the counterfactual is produced in one step. All words not included in the selected rationale from the original document, $X_M$, are kept in the counterfactual document. The kept tokens are $X \backslash X_M$. All words in the selected rationale are replaced by the CF Predictor using one prediction from the MLMs in a greedy fashion. After training, when producing the counterfactual dataset, the counterfactual documents still keep all non-rationale tokens. We now replace the rationale tokens from left to right using the output of the CF Predictor MLMs. A counterfactual token at position $t$ is decoded according to the following process.

$$x_{1,t}^c = \underset{x_{1,t}}{\arg\max}\, p(x_{1,t}|x_2, 1 - y_1, x_{1,0...t}^c) \tag{10}$$

We found this to be a good trade-off between greedy decoding and a more expensive beam search. Greedy decoding could generate frequent, repeated tokens while beam search could be an unnecessary expense for generating documents that reflect the target distribution, but do not necessarily need to pass the bar of human readers.

Figure 5 illustrates an original document and its counterfactual generated by our CF predictor. Notice here the initial selector was successful in identifying the smell aspect. The inclusion of the original and its counterfactual document in the augmented dataset successfully decreases the mutual information between the non-smell text and the smell label.

## 4 Experiments

### 4.1 Datasets

We conduct experiments using datasets from two sources. This first source contains reviews compiled by Wang et al. [29] from *TripAdvisor.com*. We use the training, dev, and test sets curated by Bao et al. [2] and used for rationalization by Chang et al. [5]. The label is binary, and we focus on the *location* aspect. There are 198 test samples with human-annotated rationales.

The second source consists of reviews collected by McAuley et al. [24] from *RateBeer*. Each review is a paragraph of text with five numerical scores in the range of 1 to 5 for the appearance, smell, palate, taste, and overall aspects of the beer. We focus on the appearance, smell, and palate aspects. We created training and dev datasets for each aspect that follow the source distribution. Following [5], we binarize this data so that all reviews with a score $\geq 3$ are class 1 and all reviews with a score $\leq 2$ are class 0. We then balance the dataset between classes. Additionally, we use procedurally decorrelated datasets created by Lei et al. [19], and binarize these datasets as well. We now have two

| Label | Document |
|---|---|
| negative | presentation : 12 oz clear bottle , label is metallic with an emblem of a buck . inkjetted on the neck i 'm not sure if this is out of date , or was produced then . appearance : pours clear , somewhere between an amber and a brown color . average head , but it fades pretty quickly to leave just a hint of lace **. smell : a bit of grain , fairly mild . this is sort of a** generic beer aroma , without much to pick out specifically . taste : a little bit of sweetness and malt this tastes a bit like a light marzen . overall impression : it is n't bad it is sort of a generic beer with a bit more flavor than your average macro sort of like a yuengling . nothing to write home about , but it 's drinkable enough . |
| positive | presentation : 12 oz clear bottle , label is metallic with an emblem of a buck . inkjetted on the neck i 'm not sure if this is out of date , or was produced then . appearance : pours clear , somewhere between an amber and a brown color . average head , but it fades pretty quickly to leave just a hint of lace **. smell : caramel malt and toffee dominate the nose with a bit of raisin and** generic beer aroma , without much to pick out specifically . taste : a little bit of sweetness and malt this tastes a bit like a light marzen . overall impression : it is n't bad it is sort of a generic beer with a bit more flavor than your average macro sort of like a yuengling . nothing to write home about , but it 's drinkable enough . |

Figure 5: A subset of the augmented dataset for beer smell. The rationale is bold in the original document (top). The replaced words are bold in the counterfactual document (bottom). $x_1$ and $y_1$ have changed from original to counterfactual. Ground truth rationales and counterfactuals are not provided in any training data.

datasets for each RateBeer aspect: a "correlated" set and a "decorrelated" set. Appendix Section A.2 shows correlation matrices before and after decorrelating the data as well as additional dataset details. In the correlated datasets, the correlations and presumably the mutual information between aspects are much higher than in the decorrelated datasets. This makes the rationalization task more difficult. Additionally, McAuley et al. [24] provides about 1,000 holdout samples where the aspect specific text is annotated by human experts.

## 4.2 Baselines

Three MMI-based baselines are used. First, "MMI" is the original Rationalization scheme [19] that has been updated and re-implemented to use the most recent NLP models as described in section 3.1. This model is trained on the original, unaugmented dataset.

Second, Factual Data Augmentation (FDA) is used to verify that our gains are not only due to data augmentation but due to the CDA procedure. FDA augments the original dataset with new samples generated by the CF predictor models, but instead of flipping the label and passing it to the $1 - y_1$ component, we pass the sample to the component of the CF predictor with the *same* label as the original document during inference. The new samples are produced using the following process:

$$y_1^c \leftarrow y_1; \quad x_1^c \leftarrow \arg\max_{x_1} p(X_1|y_1, x_2) \tag{11}$$

The third baseline, simple substitution using antonyms (ANT), does not use neural models to augment counterfactual data. The counterfactual is generated by replacing words in the rationale with antonyms from WordNet [25] [4]. Note that we only accept antonyms that are in the vocabulary used by the models and antonyms that have the same part of speech as labeled by NLTK [4].

## 4.3 Experiment Settings and Assumptions

We train the rationale selector and the classifier together, early stop based on the selector cost, freeze the selector, and finally fine-tune the classifier on the original dataset. For all of the data sets and models, we use the dev set for early stopping (more details in Appendix Section A.3). Our MLM transformers [28] were pretrained on *unlabeled* data from the TripAdvisor and RateBeer datasets *separately*. For the TripAdvisor dataset, we pretrain on all data that does not appear in the location aspect's train, dev, or annotated dataset from [5]. For the *RateBeer* datasets, we pretrain on all data that does not appear in any train, dev, or annotated dataset from [19] and the correlated datasets. We used a masking rate of 10% and masked tokens were treated as described in [9]. For the rationale models, the transformers are initialized from models pre-trained with random masking. We found

---

*Hotel - Location*

we only stayed one night  . the  location  was  great and the premises was beautiful  . the room was a typical hotel room but was decent ,  clean  and  spacious  enough for our needs . we were given a first floor room next to the pool . it was relatively quiet . the staff  were informative and  friendly . the daily parking was a total rip - off at $ 20/day but i would think that the rates are similar anywhere else you stay on the island .  all in  all  , it was a good enough stay .

*Correlated - Smell*

pours    two    thick    fingers    of    tan    froth    on    a    dark    brown    body    .    good nose , with caramel and brown sugar sweetness arising  to greet  it .  the  taste  blends  the  sweetness with a dash of hopped bitterness well .  well carbonated  , this is a sturdy , very drinkable beer that has joined me for much leisure time . i wo n't apologize for that ; nor will you after a couple brown ales .

---

Figure 6: Examples from the annotated sets. Hotel-Location (top) and Correlated-Smell (bottom). ***Human annotations are underlined***. CDA rationales are  in blue . MMI rationales are  in red . Overlaps between CDA and MMI are  in magenta .

that contiguous masking was better for pre-training the CF predictor. This contiguous masking is similar to that from Joshi et al. [16], but masked tokens are treated the same as in Devlin et al. [9] and there is only one contiguous span. The rationale selector and classifier are initialized from the random-masking transformers. For the rationale selectors, following [5], we set the rationale percentage to 10% for all datasets. The CF predictor components and the GAN discriminator are initialized from the contiguous-masking transformers. These models all have a vocabulary with $2^{15}$ tokens, 8 layers, 8 attention heads, and a hidden dimension of 256. Appendix Section A.4 shows our server configurations and more details on our experiment setup.

Models are selected and reported based on the best performance on the dev set across a grid search. All methods are evaluated using the same grid search when training the rationale model. The model is selected before fine-tuning the classifier with a frozen rationale selector. The initial selector used to train the counterfactual predictors was the selected MMI model with a fine-tuned classifier. The parameters and checkpoints for the CF Predictor models are tuned and chosen to maximize the accuracy of the training documents' predicted label as compared to the target label (measured by the original rationale model) and to maximize the entropy in the inserted counterfactual tokens. The CF Predictor model is chosen from a grid search, using only the training dataset, across $\lambda_A$ and $\lambda_{RL}$.

For the TripAdvisor dataset, we repeat the experiment three times: training an initial rationale selector, training a counterfactual predictor, generating a counterfactual dataset, and training a new rationale model. Additionally, we train two additional rationale models with different random seeds and the selected hyperparameters. Consequently, there are nine rationale models for each of the four methods, and we report the mean and standard deviation across those nine models. In the RateBeer datasets, we repeated the same experiments two times and then trained two additional models with different random seeds and the selected hyperparameters for each rationale model. As a result, we have a total of six runs for each method for the RateBeer datasets. All models are in Tensorflow [1]. Our software is publicly released [1].

### 4.4 Results

We evaluate the rationale models by the precision (Rat. Prec.) as compared to human annotations. This token-level metric is taken as the mean across samples in the annotated set. We also report the accuracy of the classifier on the development set (Dev. Acc.).

As shown in Tables 1, 2, and 3, our CDA approach outperforms all baselines on $\frac{6}{7}$ experiments. As expected, CDA's performance over other methods is most pronounced on the correlated RateBeer datasets compared to on the decorrelated ones. Only on the decorrelated appearance dataset, CDA performs second to MMI. This may be because on the decorrelated appearance dataset the spurious aspects have very low mutual information with the target label, appearance, and therefore the error margin for the initial selector is very tight and CDA is expected to yield low gains as discussed in Section 2.4.

---

[1]github.com/mlplyler/CFs_for_Rationales

The baseline augmentation schemes (FDA, ANT) produced mixed results. The FDA scheme should not have changed the mutual information between the spurious aspects and the target label. Whenever it performed worse than MMI alone, it most likely introduced noise into the dataset. While the ANT scheme might be effective in lowering the mutual information between the spurious aspects and the target label, the generated counterfactual does not likely model $p(x_1|1 - y_1, x_2)$, so we expect it to also reduce $I(X_1, Y_1)$.

Empirically, we have demonstrated that models trained on the CDA augmented data tend to outperform models trained on the original datasets. This lends credence to the idea that our scheme is indeed lowering the mutual information between the spurious aspects and the target label. The dataset and method pairs with very high variance are dragged down by degenerated runs where the rationale selector has little or no skill. These runs were seen when varying the random seed with the selected hyperparameters. Removing the degenerate runs can bring the mean rationale precision of the high variance experiments closer to CDA, but we believe these degenerate models seen primarily in the baselines align well with our theory that CDA allows the rationale selector to more easily identify $X_1$.

Table 1: TripAdvisor - Location

|     | Dev. Acc. | Rat. Prec. |
| --- | --- | --- |
| MMI | $78.16 \pm 5.83$ | $26.14 \pm 13.25$ |
| FDA | $80.61 \pm 4.38$ | $31.36 \pm 10.38$ |
| ANT | $69.79 \pm 2.90$ | $12.25 \pm 3.64$ |
| CDA | $78.11 \pm 6.77$ | $\mathbf{39.76} \pm 10.48$ |

**Case Study:** Figure 6 presents a case study comparing the rationales selected by MMI and our proposed CDA approach in the TripAdvisor and Correlated Beer datasets respectively. As shown in Figure 6, our CDA models can select the text that aligns better with human annotations and they successfully avoid selecting sentiment-carrying text that is not relevant to the aspect of interest.

Table 2: Correlated RateBeer Results

|     | Appearance | | Smell | | Palate | |
| --- | --- | --- | --- | --- | --- | --- |
|     | Dev. Acc. | Rat. Prec. | Dev. Acc. | Rat. Prec. | Dev. Acc. | Rat. Prec. |
| MMI | $76.37 \pm 3.05$ | $56.28 \pm 17.22$ | $79.49 \pm 4.53$ | $45.39 \pm 16.45$ | $82.67 \pm 0.99$ | $34.98 \pm 8.79$ |
| FDA | $75.73 \pm 3.64$ | $41.02 \pm 5.09$ | $76.43 \pm 5.08$ | $41.67 \pm 23.73$ | $77.60 \pm 6.06$ | $32.35 \pm 14.41$ |
| ANT | $62.09 \pm 8.76$ | $43.25 \pm 16.51$ | $57.70 \pm 9.87$ | $11.72 \pm 7.16$ | $61.93 \pm 7.59$ | $8.62 \pm 12.31$ |
| CDA | $73.30 \pm 2.93$ | $\mathbf{67.79} \pm 10.63$ | $81.82 \pm 0.69$ | $\mathbf{61.84} \pm 6.76$ | $80.65 \pm 0.64$ | $\mathbf{41.24} \pm 1.72$ |

Table 3: Decorrelated RateBeer Results

|     | Appearance | | Smell | | Palate | |
| --- | --- | --- | --- | --- | --- | --- |
|     | Dev. Acc. | Rat. Prec. | Dev. Acc. | Rat. Prec. | Dev. Acc. | Rat. Prec. |
| MMI | $81.33 \pm 0.93$ | $\mathbf{86.92} \pm 7.35$ | $77.99 \pm 5.46$ | $79.71 \pm 10.56$ | $68.64 \pm 10.09$ | $46.39 \pm 29.21$ |
| FDA | $80.67 \pm 1.32$ | $81.97 \pm 6.00$ | $79.66 \pm 1.23$ | $83.62 \pm 3.52$ | $77.26 \pm 1.90$ | $66.03 \pm 3.76$ |
| ANT | $71.41 \pm 9.97$ | $67.67 \pm 27.68$ | $58.81 \pm 6.31$ | $9.32 \pm 10.76$ | $57.75 \pm 7.98$ | $10.83 \pm 11.15$ |
| CDA | $80.24 \pm 1.07$ | $82.82 \pm 8.60$ | $79.18 \pm 1.43$ | $\mathbf{86.66} \pm 2.92$ | $76.81 \pm 0.85$ | $\mathbf{67.79} \pm 3.20$ |

## 5    Related Work

The rationale framework used in this work was introduced in [19] and connected to the MMI criteria by Chen et al. [7]. Much of the follow-up work introduced modifications to the learning algorithm to overcome spurious signals picked up by MMI. Yu et al. [33] sought to limit the signal left in the complement of the rationales. The concept of class-wise rationalization was introduced in Chang et al. [5], and Chang et al. [6] introduced a rationale algorithm that seeks invariant rationales across environments. These works modify the rationale framework and the MMI criteria to help the rationale selector find desirable signals. In our work, we keep the MMI criteria unchanged, but instead seek to diminish the undesirable signals through counterfactual data augmentation.

Data augmentation generally and counterfactual data augmentation (CDA) specifically has been a popular technique in recent natural language processing (NLP) work. Lu et al. [22] introduced counterfactual data augmentation as a general methodology and showed their rule-based scheme

could mitigate gender biases commonly seen in word embeddings and NLP models. Other works used CDA to train better named entity recognition (NER) models in the medical text domain [34]. More work [17] continued the ideas of CDA by working with sentiment analysis and natural language inference tasks. They constructed their CDA datasets using crowd workers and showed models trained on the augmented datasets generalized better out-of-domain. Follow-up work [18] showed how their crowd worker interventions reduce spurious signals when viewed through a simplified causal modeling lens. The causal lens and generative counterfactual data augmentation have also shown recent, positive results in the computer vision domain [27].

# 6 Conclusions

This work presents a counterfactual data augmentation method for lowering the mutual information between spurious signals and a target label in a dataset. We derive theoretical conditions that indicate when our approach will be beneficial to the dataset. Empirically, we show rationale models trained with MMI on our counterfactually augmented datasets result in rationale selectors with improved behaviours. Furthermore, our analysis shows that the benefits of our scheme are proportional to the error rate of the original rationale selector. This suggests we could perform counterfactual data augmentation iteratively where we achieve better rationale selectors after each iteration that could be used as the initial selector in the next round of data augmentation.

Our method does not make any changes to the core rationale algorithm. There has been more work in the literature on rationale algorithms since Lei et al. [19] and Chen et al. [7], but they were not included in this study because they do not necessarily follow the maximum mutual information criteria on which our theory is based. From a practical standpoint, it is likely our CDA method could connect with other rationalization strategies because the CF Predictor is trained using a frozen rationale model. This could be an important area for future study.

# 7 Broader Impact

This work focuses on methods for training interpretable NLP models. While interpretability is typically a desired trait, defining exactly what is interpretable is not always clear [21]. Rationale models have been shown to not always be faithful [14] [13], and in the worst case, they can degenerate in such a way where the rationale selector makes the decision instead of the classifier [33]. The degenerated case is likely to produce rationales that are nonsensical to human interpretation.

Section 2.4 presents some theoretical proof on how CDA affects the mutual information between text and the target label with different initial selector error rates. Some conclusions drawn in Section 2.4 rely on a simplified model with binary variables. Furthermore, the simplified model makes the assumption that the generated text follows the same distribution as the original dataset. While this is the goal of the adversarial learning strategy that we implemented, we found that the generated text had lower entropy than the rationales they replaced. This is likely due to both the $\arg\max$ in Eqn 2 and $L_{RL}$.

We have demonstrated positive results for improving rationales for sentiment analysis sentiment classifiers. Generalizing the approach to other NLP tasks is less straightforward. First, the naive approach to generalizing to $K$ classes in the classification setting suggests having $K$ CF Predictor models and generating $K-1$ counterfactual documents per data sample. This would be a significant increase in computation and resources. Second, for the task of natural language inference, CDA has been successful with human annotators [17]. It might be possible to train a CF Predictor for each type of hypothesis. It is less clear how to train a CF Predictor to modify a premise.

Our method uses data augmentation to lower the mutual information between some signals in documents and a label of interest. When the initial rationale selector aligns with human judgment, models trained on the augmented data tend to better align with human judgement. The method can be thought of as amplifying signals relative to others. We are changing the properties of the dataset. For sensitive tasks, one should check for undesired biases and fairness concerns before and after CDA.

**Acknowledgments:** We sincerely thank Jascha Swisher and Stephen Shauger for their insights and help. We thank the Laboratory for Analytic Sciences for supporting the work, and we also thank the reviewers for their time and helpful advice.

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
