# A Appendix

## A.1 Additional Rationale Results

Here we show some additional metrics for the rationale models. The recall and F1 scores shown in Tables 4, 5, and 6 taken as the mean score across documents.

Table 4: TripAdvisor - Location

|  | Rat. F1 | Rat. Recall |
|---|---|---|
| MMI | 24.89 $\pm$ 13.16 | 30.96 $\pm$ 16.81 |
| FDA | 30.07 $\pm$ 10.45 | 37.44 $\pm$ 13.29 |
| ANT | 11.80 $\pm$ 3.59 | 15.61 $\pm$ 4.84 |
| CDA | **38.29** $\pm$ 10.40 | **47.38** $\pm$ 13.07 |

Table 5: Additional Decorrelated RateBeer Results

|  | Appearance | | Smell | | Palate | |
|---|---|---|---|---|---|---|
|  | Rat. F1 | Rat. Recall | Rat. F1 | Rat. Recall | Rat. F1 | Rat. Recall |
| MMI | **62.36** $\pm$ 5.41 | **51.82** $\pm$ 4.58 | 63.84 $\pm$ 8.73 | 57.25 $\pm$ 7.96 | 41.97 $\pm$ 26.85 | 42.80 $\pm$ 27.69 |
| FDA | 58.37 $\pm$ 4.72 | 48.26 $\pm$ 4.20 | 67.06 $\pm$ 2.94 | 60.24 $\pm$ 2.69 | 59.96 $\pm$ 3.41 | 61.26 $\pm$ 3.47 |
| ANT | 47.81 $\pm$ 20.36 | 39.31 $\pm$ 17.11 | 7.17 $\pm$ 8.02 | 6.28 $\pm$ 6.78 | 9.27 $\pm$ 9.90 | 9.06 $\pm$ 9.85 |
| CDA | 59.13 $\pm$ 6.55 | 48.96 $\pm$ 5.69 | **69.56** $\pm$ 2.44 | **62.45** $\pm$ 2.26 | **61.59** $\pm$ 2.82 | **62.96** $\pm$ 2.77 |

Table 6: Additional Correlated RateBeer Results

|  | Appearance | | Smell | | Palate | |
|---|---|---|---|---|---|---|
|  | Rat. F1 | Rat. Recall | Rat. F1 | Rat. Recall | Rat. F1 | Rat. Recall |
| MMI | 39.44 $\pm$ 12.75 | 32.32 $\pm$ 10.76 | 35.66 $\pm$ 13.05 | 31.62 $\pm$ 11.61 | 31.34 $\pm$ 8.03 | 31.85 $\pm$ 8.18 |
| FDA | 27.70 $\pm$ 3.81 | 22.22 $\pm$ 3.23 | 32.72 $\pm$ 18.93 | 28.96 $\pm$ 16.91 | 29.31 $\pm$ 13.12 | 30.03 $\pm$ 13.46 |
| ANT | 30.03 $\pm$ 11.96 | 24.49 $\pm$ 9.99 | 9.14 $\pm$ 5.51 | 8.05 $\pm$ 4.76 | 7.57 $\pm$ 11.03 | 7.56 $\pm$ 11.10 |
| CDA | **48.02** $\pm$ 7.64 | **39.56** $\pm$ 6.28 | **48.98** $\pm$ 5.56 | **43.68** $\pm$ 5.12 | **37.33** $\pm$ 1.64 | **38.33** $\pm$ 1.87 |

## A.2 Dataset Descriptions

Table 7 describes the size of the datasets used in this work. For the *RateBeer* datasets, the correlated and decorrelated datasets are the same size for each of the three aspects. After binarizing the data, the data is balanced by taking the samples in the order that they appear in the source files. We do not include annotated samples with empty rationales. For the TripAdvisor dataset, the training dataset is modified from its source such that no dev or test samples are in the training set. All documents are a maximum of 256 tokens.

Figure 7 shows correlation matrices for the decorrelated (top) and correlated (bottom) datasets for *RateBeer*. This is for non-binarized data but after balancing the classes. The correlated datasets more closely follow the distribution of the full *RateBeer* datasets. Because there is much more correlation between the aspects, the rationale selector has a more difficult time identifying the correct aspect. To our knowledge, these datasets are open for academic use.

## A.3 Training Procedure

### A.3.1 Masked-Language-Models

We trained MLMs for the *RateBeer* and *TripAdvisor* data separately. These MLMs used a dropout rate of 10%. The *RateBeer* random-mask and contiguous-mask MLMs used a batch size of 128 across 4 GPUs and trained for one week. The *TripAdvisor* random-mask MLM used a batch size 256 across 8 GPUs while the contiguous-mask MLM used a batch size of 128 across 4 GPUs. Both trained for one week.

Table 7: Datasets Descriptions. Formatted as class-0 / class-1.

| Source | Dataset | Train | Dev | Annotated |
|---|---|---|---|---|
| TripAdvisor | Location | 6,715 / 6,715 | 895 / 895 | 95 / 103 |
| RateBeer (Decorr) | Appearance | 16,891 / 16,891 | 2,103 / 2,103 | 13 / 923 |
| | Smell | 15,169 / 15,169 | 2,218 / 2,218 | 29 / 848 |
| | Palate | 13,652 / 13,652 | 2,000 / 2,000 | 20 / 785 |
| RateBeer (Corr) | Appearance | 16,891 / 16,891 | 2,000 / 2,000 | 13 / 923 |
| | Smell | 15,169 / 15,169 | 2,000 / 2,000 | 29 / 848 |
| | Palate | 13,652 / 13,652 | 2,000 / 2,000 | 20 / 785 |

Figure 7: RateBeer Datasets Correlations. Decorrelated (top) and Correlated (bottom).

### A.3.2 Rationale Models

We train the rationale models in two phases. For the first, we train the rationale selector and the classifier together. We train for 21 epochs and early stop based on the selector cost. The second phase is for training the classifier only on the original dataset. This is done to recover any loss in accuracy due to noise introduced by data augmentation, but we do this for the models trained on the original data only as well. This second stage is only done for the selected models.

Across a grid search for a method and dataset pair, we select the model with the lowest $L_s$. Here, this is defined as $L_s = \lambda_c L_c + \lambda_y L_y$. $\lambda_c$ and $\lambda_y$ are taken as the inverse means of $L_s$ and $L_c$ across the grid search.

We initialized a grid search for all datasets and methods over the coherent regularizer $L_c$, and we perform this grid search in two stages. The first stage $L_c$ values considered for *RateBeer* datasets were $[5, 10, 15, 20]$ and the learning rate was 1e-4 for all models. The second stage $L_c$ values used were $[L_{c1} - 2, L_{c1} - 1, L_{c1} + 1, L_{c1} + 2]$ where $L_{c1}$ is the best $L_c$ value from the previouus stage. Each run across all searches used a uniquely drawn random seed, a batch size of 50, a dropout rate of 5% for the selector, and a dropout rate of 10% for the classifier. The rationale classifier used max-out across the time dimension. The rationale selector and the classifier were trained using the Adam optimizer. Table 8 shows the final selected hyper-parameters for all reported rationale models.

### A.3.3 CF Predictor Models

The CF predictor models were trained using only the original training dataset. Hyper-parameters $L_A$ and $L_{RL}$ were tuned and checkpoints were chosen based on the percentage of labels that were successfully flipped as measured by the rationale model and based on the entropy of the infilled

| Dataset/Experiment | MMI | FDA | ANT | CDA |
|---|---|---|---|---|
| Location 1 | 8 | 15 | 10 | 12 |
| Location 2 | 20 | 18 | 20 | 5 |
| Location 3 | 17 | 17 | 12 | 16 |
| Appearance-Decorr 1 | 17 | 8 | 5 | 21 |
| Appearance-Decorr 2 | 14 | 15 | 5 | 21 |
| Smell-Decorr 1 | 18 | 22 | 16 | 15 |
| Smell-Decorr 2 | 20 | 20 | 20 | 22 |
| Palate-Decorr 1 | 20 | 20 | 5 | 22 |
| Palate-Decorr 2 | 15 | 20 | 17 | 22 |
| Appearance-Corr 1 | 12 | 16 | 9 | 9 |
| Appearance-Corr 2 | 18 | 17 | 15 | 18 |
| Smell-Corr 1 | 20 | 15 | 10 | 18 |
| Smell-Corr 2 | 21 | 17 | 13 | 15 |
| Palate-Corr 1 | 4 | 15 | 19 | 15 |
| Palate-Corr 2 | 14 | 16 | 17 | 18 |

Table 8: Coherency lambda, $L_c$, selected for rationale models. The dataset number denotes the experiment number.

| Dataset/Experiment | $+\lambda_{RL}$ | $+\lambda_A$ | $-\lambda_{RL}$ | $-\lambda_A$ |
|---|---|---|---|---|
| Location 1 | 5 | 20 | 5 | 15 |
| Location 2 | 1 | 5 | 1 | 1 |
| Location 3 | 15 | 20 | 1 | 10 |
| Appearance-Decorr 1 | 1 | 5 | 5 | 20 |
| Appearance-Decorr 2 | 5 | 15 | 5 | 15 |
| Smell-Decorr 1 | 10 | 20 | 5 | 15 |
| Smell-Decorr 2 | 5 | 5 | 5 | 20 |
| Palate-Decorr 1 | 1 | 5 | 1 | 10 |
| Palate-Decorr 2 | 5 | 15 | 1 | 10 |
| Appearance-Corr 1 | 1 | 5 | 5 | 15 |
| Appearance-Corr 2 | 1 | 5 | 5 | 20 |
| Smell-Corr 1 | 10 | 20 | 1 | 10 |
| Smell-Corr 2 | 1 | 5 | 10 | 20 |
| Palate-Corr 1 | 1 | 20 | 1 | 5 |
| Palate-Corr 2 | 1 | 10 | 1 | 15 |

Table 9: CF Predictor Selected Hyperparameters.

words. Higher entropy is better as low entropy suggests the model is frequently infilling repeated words. CF Predictors models and checkpoints were chosen based on the metric $4.5a + t$ where $a$ is the percentage of documents whose labeled was flipped (as measured by the rationale model) and $t$ is the entropy of the infilled words. This metric was estimated using the first 500 samples in the training dataset and using one step decoding. This is not the iterative decoding method used to produce the final rationale model. We check this metric every 50 training steps. CF predictors were trained with the Adam optimizer with a linearly increasing and decreasing rate. The peak learning rate was reached in 100 steps. These models used a dropout rate of 10%.

The positive and negative CF Predictors were selected from grid searches using $[1, 5, 10, 15, 20, 25]$ for both $L_A$ and $L_{RL}$ with the condition that $L_{RL} \leq L_A$. All models were trained with the Adam optimizer with a a linearly increasing and decreasing learning rate that peaked at 5e-6 after 100 steps, 10% dropout, and for seven epochs or until the compute node failed. Table 9 shows the final selected hyperparameters.

## A.4 Server Configuration

This work was computed on a SLURM cluster with a variety of compute nodes. TripAdvisor rationale models were trained on AMD Epyc Rome CPUs and rtx2060super GPUs. The RateBeer rationale models were trained on Intel Skylake CPUs and p4000 GPUs. All CF Predictor models were trained on rtx2060super GPUs. All rationale models' classifier were finetuned on the p4000 nodes with Skylake CPUs. rtx2060, rtx2060super, and p4000 nodes were used to pre-train the MLMs.