# OpenReview forum: "Making a (Counterfactual) Difference One Rationale at a Time"
_NeurIPS.cc/2021/Conference — NeurIPS 2021 Poster_

### Official Review · Reviewer_5Z3P · 2021-07-05

**Rating:** 8
**Confidence:** 4

**Summary:**

A rationale in NLP is a subset of an original input text on which a classifier operates. Extracting such a rationale is a popular approach for more interpretability because by definition we know the classifier only had access to this reduced text. However, rationales can be nonsensical and can consist of spurious patterns in the dataset.

To improve the coherence of rationales and to improve generalization, this paper proposes a technique to automatically create counterfactual data. This counterfactual data is created specifically to reduce mutual information between the true target label and irrelevant parts of the original text with regards to the true target label. For example, in “This beer smells great. It tastes terrific.” both sentences have positive sentiment but only the first is relevant when predicting whether the beer has a good smell. To help a model to automatically learn to distinguish, a helpful counterfactual example would be “This beer smells awful. It tastes terrific”.

Contributions
-	A method to automatically generate counterfactual examples to improve rationale extraction.
-	An analysis of how good an initial rationale extractor should be to be able to expect performance improvements using the proposed method of generating counterfactual examples.


**Limitations And Societal Impact:**

There is a broader impact section which discusses the limitations and dangers adequately.

**Main Review:**

Strengths
-	The proposed idea is intuitive and interesting.
-	Experimental results show improvements over baselines (but also see point 2 under weaknesses).

Weaknesses
-	Some parts of the paper are difficult to follow, see also Typos etc below.
-	Ideally other baselines would also be included, such as the other works discussed in related work [29, 5, 6].

After the Authors' Response
My weakness points after been addressed in the authors' response. Consequently I raised my score.
- All unclear parts have been answered
- The authors' explained why the chosen baseline makes the most sense. It would be great if this is added to the final version of the paper.

Questions
-	Do you think there is a way to test beforehand whether I(X_1, Y_1) would be lowered more than I(X_2, Y_1) beforehand?
-	Out of curiosity, did you consider first using Aug and then CF.CDA? Especially for the correlated palate result it could be interesting to see if now CF.CDA can improve.
-	Did both CDA and MMI have the same lambda_RL (Eq 9) value? From Figure 6 it seems the biggest difference between CDA and MMI is that MMI has more discontinuous phrase/tokens.

Typos, representation etc.
-	Line 69: Is X_2 defined as all features of X not in X_1? Stating this explicitly would be great.
-	Line 88: What ideas exactly do you take from [19] and how does your approach differ?
-	Eq 2: Does this mean Y is a value in [0, 1] for two possible labels? Can this be extended to more labels? This should be clarified.
-	262: What are the possible Y values for TripAdvisor’s location aspect?
-	The definitions and usage of the various variables is sometimes difficult to follow. E.g. What exactly is the definition of X_2? (see also first point above). When does X_M become X_1? Sometimes the augmented data has a superscript, sometimes it does not. In line 131 the meaning of x_1 and x_2 are reverse, which can get confusing - maybe x’_1 and x’_2 would make it easier to follow together with a table that explains the meaning of different variables?
-	Section 2.3: Before line 116 mentioned the change when adding the counterfactual example, it would be helpful to first state what I(X_2, Y_1) and I(X_1, Y_1) are without it.

Minor points
-	Line 29: How is desired relationship between input text and target labels defined?
-	Line 44: What is meant by the initial rationale selector is perfect? It seems if it were perfect no additional work needs to be done.
-	Line 14, 47: A brief explanation of “multi-aspect” would be helpful
-	Figure 1: Subscripts s and t should be 1 and 2?
-	184: Delete “the”

**Time Spent Reviewing:**

3

---

> ### Author Response · Authors · 2021-08-10
> **Response to reviewer 5Z3P**
>
> We thank the reviewer for their time and detailed review.
>
> [Comment:] Ideally other baselines would also be included, such as the other works discussed in related work [29, 5, 6] \
> [Response:] We chose the MMI-based baselines because it best matched our theoretical findings that our method had the ability to lower the mutual information between spurious and desired signals. In our view, our CDA method could also benefit other rationale extraction methods since spurious signals are a common problem for all of them. The idea could be considered for future work. Specifically, Invariant Rationalization was not directly compared because it requires the training data to be split into environments, and this is not a presupposition we make.
>
> [Comments:] Some parts of the paper are difficult to follow, see also Typos etc below. \
> [Responses:] If accepted, we will make the following changes. \
> [Line 69]  In future versions, we will make this clearer: X_2 is all features that are not included in X_1. It follows that there is some text that represents the desired relationship or even cause-and-effect relationship and then there is all the rest. \
> Line 88: In this study, we considered binary values of Y, so yes values [0,1]. It holds true both for the RateBeer dataset and for the TripAdvisor dataset. We believe that the method could be generalized to more classes. In a naive approach, each class would be given a counterfactual. If a sample had class k out of a set K of classes, you could generate K-1 counterfactuals for all other classes. This should balance X_2 across all other classes.
>
> [Other presentation and Minor points questions:] \
> A multi-aspect dataset is one in which multiple characteristics, or aspects, are discussed in the same document. A beer review includes discussion of smell, taste, and smell for the beer. In addition, these datasets commonly have labels for different aspects, and scores for every aspect separately - but these aspect labels are not necessary or used in our method.
> A dataset's and a task's characteristics determine the ideal relationship between input text and target labels. As with the multi-aspect datasets, a rationale text describing an aspect label is ideal. The RateBeer dataset makes an excellent demonstration dataset since a subset of text talks clearly about the smell of the beer, for instance.
> A perfect rationale selector is one that always identifies the "ground truth" or correct rationale. You are correct that if the initial selector was perfect then no work needs to be done. We started with the analysis of the perfect initial selector to show that in the limit, our method would perfectly eliminate the mutual information between the spurious text and the target label. The perfect initial selector is not realistic and as you pointed out, makes our method unnecessary, so we continued our analysis with noisy initial selectors.
> In Figure 1,  subscripts s and t can are 1 and 2. We tried to make the figure stand alone and mapped smell \rightarrow s and taste \rightarrow t, but we will seek to make this clearer in future versions.
> We thank-you for identifying the typos and latex mistakes and we will fix those in future versions.
>
> [Question:] Do you think there is a way to test beforehand whether I(X_1, Y_1) would be lowered more than I(X_2, Y_1)beforehand? \
> [Response:] The test of whether I (X_1, Y_1) would be lowered more than I (X_2, Y_1) might require knowledge of rationale quality a priori, which we cannot assume since "ground truth" labels are only provided in the test dataset. One might be able to assess the rationales in a cruder fashion without using rationale annotations by analyzing tokens and bigrams selected by the initial rationale selection. Taking Figure 3, if the rationale selector includes "no lacing" more often than it includes "light bodied", the initial selector is probably on the right track, so I(X_1,Y_1) would be lowered more than I(X_2,Y_1). This approach obviously would require domain knowledge.
>
> [Question:] Out of curiosity, did you consider first using Aug and then CF.CDA?  \
> [Response:] We did not consider this, but using both CDA and Factual data augmentation is a fantastic suggestion. Thank you! Factual data augmentation did outperform MMI for 6/7 datasets, so it likely would have been a better initial selector than the MMI model. We will consider this for future work.
>
> [Question:] Did both CDA and MMI have the same lambda_RL (Eq 9) value? \
> [Response:]  lambda_RL is not used during MMI training. In Figure 6, the distinction between rationale predictor training and counterfactual predictor training appears unclear. Upon acceptance, we will clarify this in the final version: All rationale models include the selector and the classifier, and are trained using only those two modules. Between CDA and MMI, the former is trained on an augmented dataset, while the latter is trained only on the original dataset only.

---

### Official Review · Reviewer_Sfbi · 2021-07-15

**Rating:** 7
**Confidence:** 3

**Summary:**

This paper aims to provide a novel rationalization scheme for text classification models in NLP. The authors use the standard setup of a selector and classifier model wherein the selectors selects specific words from the input text and classifier makes a prediction on the selected rationale. Standard issues in such models is that the selector may end up picking on spurious correlations (or "artifacts") when picking the rationale if they are correlated with the predicted quantity, instead of true predictive features. The core contribution of the authors is to augment the dataset in such a manner that the correlation between the artifacts and the label is broken.

The correlation is broken by generating counterfactual examples -- examples where predictive features are *flipped* along with the label but the spurious features are kept the same. Since we don't know a priori which features are predictive vs spurious, the authors hope that the dataset contains slightly less information about the label than the predictive features and provide an analysis in a simple binary feature case, wherein the show when the model may increase the mutual information for predictive features vs rationale error rate of initial selector.

The modeling part use transformer based models all the way. The setup learns the selector and prediction, in addition to a counterfactual generator (which replaces the rationale words with words predictive of the opposite class), and a discriminator to make sure the words are indeed from same distribution as input text.

Experiments are done on two sentiment analysis multi-aspect datasets and the generated rationales are shown to be more accurate than baseline MMI training. Some qualitative analysis seem to indicate better rationales are produced in terms of coherency and correctness.


**Limitations And Societal Impact:**

Please include a bit more talk about how you believe the method would generalize beyond simple sentiment analysis (for example, NLI, etc)

**Main Review:**

While both rationalization and counterfactual augmentation have been performed before, this paper provides first known implementation to correcting for spurious correlation via counterfactuals. The authors provide relatively well researched section for related work although primarily focuses on MMI based rationalization (https://direct.mit.edu/tacl/article/doi/10.1162/tacl_a_00367/98620/Aligning-Faithful-Interpretations-with-their) . In addition, comparisons to other baselines like Invariant Rationalization has not been made (It is unclear why that is).

The basic idea behind the paper is reasonably clear from the first reading, although some details can be improved. For example, in line 146, it is unclear what $X^{er}_1$ is. In addition, equations 5 and 6 are not immediately obvious from preceding discussion and can use a few more lines of derivation. Also it seems the derivation assumes that predicted label is binary (the counterfactual label is 1 - Y). What if the tasks is multi class -- how will the derivation generalize ? The details for experiments are reasonably sufficient that I can reproduce the results (assuming the code is also provided).

In terms of significance, the spurious correlation problem is of course an issue in rationalization literature with not many solutions. the authors provide a novel strategy to solve the same and the results in terms of rationale accuracy is better compared to standard training. Is the improvement because the model is better at not picking up the spurious correlation -- I don't know. An interesting experiment would be to see if the model trained with MMI pick up on rationales from other aspect more than the specific aspect in the correlated setting.

The biggest issue with the paper appears to be that the experiments are mostly run on two datasets, both belonging to sentiment analysis domain only. While results show improvements in these two datasets, it is unclear if they will generalize beyond these datasets or beyond sentiment analysis domain.

**Time Spent Reviewing:**

3

---

> ### Author Response · Authors · 2021-08-10
> **Response to reviewer Sfbi**
>
> We thank the reviewer for their time and detailed review.
>
> [Comment:] The need to compare to other baselines including Invariant Rationalization \
> [Response:] We chose the MMI-based baselines because it best matched our theoretical findings that our method had the ability to lower the mutual information between spurious and desired signals. In our view, our CDA method could also benefit other rationale extraction methods since spurious signals are a common problem for all of them. The idea could be considered for future work. Specifically, Invariant Rationalization was not directly compared because it requires the training data to be split into environments, and this is not a presupposition we make.
>
> [Comment:] Therotical Details questions \
> [Response:] X_1^{er} is the X distribution that is populated by the counterfactual predictor when the rationale selector makes a mistake in selecting spurious text. We make the assumption that this follows the same distribution as original X_1. This will be clarified in future versions.
>
> [Comment:] how would the proposed framework work on multi-class Task \
> [Response:] Due to the fact that our framework is MMI-based, we expect that our framework would still be able to handle multi-class tasks. A naive approach to extending beyond the two class (binary label) setting is to create a counterfactual for all other classes for each sample. In cases with K classes, generating K-1 counterfactual documents would distribute the 'non-causal' information across those classes. Obviously, this is an expensive proposition, and we can look into more efficient methods as part of future research. Last but not least, our code has been submitted.  We plan to publicly release code upon acceptance as well.
>
> [Comment:] Should try the proposed work on other Datasets. \
> [Response:] The datasets considered in this work are sourced from two larger datasets. While our experiments are based on TripAdvisor and RateBeer, please note that there are actually 7 different datasets we examined. For each aspect of RateBeer, we present data for two versions, correlated and decorrelated. These are standard datasets found as part of the rationale literature, for instance [5, 6, 16].
>
> [Comment:] Limitations And Societal Impact should include NLI. \
> [Response:] Thanks for pointing it out. In order to generalize the method to NLI, we would first compare our method to the manually crafted counterfactuals from [14]. In that work, they used crowd workers to craft counterfactuals that changed either the premise or hypothesis. It might be possible for us to have a counterfactual predictor for each type of hypothesis. Specifically, there would be one model that transfers entailment and contradiction documents into neutral. There would be two more models that generate entailment and contradiction documents. For each point in the original datasets, we would generate counterfactual documents for the two other hypothesis labels. This would ideally balance the spurious information across the hypothesis labels. A similar approach might be possible for the premises.

---

> > ### Comment · Reviewer_Sfbi · 2021-08-18
> > **Dataset Issues**
> >
> > Thanks for the response. I am not sure what are the 7 datasets considered are -- from the paper and Supp material, it seems all experiments were run on TripAdvisor and RateBeer Datasets (with different aspects or correlation structures). Moreover, all 3 papers [5, 6, 16] you mention are from Tommi Jakkola's group at MIT, hence I won't consider them as standard in the field (for example, other rationale type papers from https://cs.jhu.edu/~jason/papers/zaidan+al.nipsw08.pdf , https://arxiv.org/abs/2005.00115 , https://direct.mit.edu/tacl/article/doi/10.1162/tacl_a_00367/98620/Aligning-Faithful-Interpretations-with-their , etc).

---

> > > ### Author Response · Authors · 2021-08-19
> > > **Response to Dataset Issues**
> > >
> > > Thank-you again for your time, questions, and comments.
> > >
> > > [Question] I am not sure what are the 7 datasets considered are -- from the paper and Supp material, it seems all experiments were run on TripAdvisor and RateBeer Datasets (with different aspects or correlation structures). \
> > > [Response:] We will be more clear in future versions that we used data from two sources with multi-aspect labels: RateBeer and TripAdvisor.  A total of seven datasets were extracted from the two sources: the TripAdvisor "location" aspect, and three aspects from RateBeer using correlated settings, and three aspects from RateBeer using decorrelated settings. These are seven different, individual files/datasets that have been used to evaluate the performance of the proposed methods.
> > >
> > >
> > > [Comments] Moreover, all 3 papers [5, 6, 16] you mention are from Tommi Jakkola's group at MIT, hence I won't consider them as standard in the field \
> > > [Response:] We completely agree with your assessment of our references in our rebuttal. There is no intent here to suggest that any one group should set the standard. Thank you for providing the additional references; these papers provide us with a great resource for other rationale datasets. Our reasoning for choosing our datasets will be clarified in future versions as follows:
> > > Considering the nature of our proposed work, we used only multi-aspect datasets and did not make use of the single-aspect datasets used in https://cs.jhu.edu/~jason/papers/zaidan+al.nipsw08.pdf , https://arxiv.org/abs/2005.00115 ).  Our work's primary motivation to ignore spurious properties in training data is most evident in the multi-aspect setting, where all other aspects are treated as spurious. Moreover, using the labels for the spurious aspects, one can compute a proxy measure for the mutual information between the target label and its spurious aspects.

---

### Official Review · Reviewer_Z6bp · 2021-07-16

**Rating:** 7
**Confidence:** 5

**Summary:**

In this paper, the authors discuss selecting rationales to improve classifier performance in an extract then classify setting. The motivation behind this paper is that prior work on rationale selection often seeks to identify a subset of tokens in a document that carry the most information about the target label. However, oftentimes such rationale selectors pick up on spurious correlations that are merely correlated with the label due to confounding and do not express the desired relationship with the label. So even though the desired predictor would rely on “the causal variable”, in this case, predictors often end up relying on the “non causal” factors.

To address this issue, the authors suggest assisting the rationale selector by lowering the mutual information between non causal features and the label (since a rationale selector will try to identify features that have the maximum mutual information with the label). To do so, the authors propose a framework in which they first use a rationale extractor (similar to that introduced by Lei et al.) to identify rationales for the label of interest. A class conditional masked language model then makes inference on these tokens (while keeping the other tokens constant) to produce new documents with a counterfactual class label. These replacements are performed by the MLM in one step (in a greedy fashion). A discriminator seeks to distinguish between the original document and the generated counterfactual document. This process is run iteratively until convergence. The core idea here is that when the original dataset will be augmented with the counterfactual documents, it should lead to a lower mutual information between the “non rationales” that remained unchanged during counterfactual generation and the label. As the authors recognize, one drawback of automatically generating counterfactual examples is that we assume access to a perfect rationale selector. However, since the motivation of the paper relies on a poor rationale selector which may also reduce the mutual information between causal features and the label, they conjecture that reducing the mutual information between spurious features and the label more than reducing mutual information between causal features and the label could allow us to improve the model performance. The authors conduct experiments on Beer reviews from RateBeer and Tripadvisor reviews. Compared to three baselines, the proposed method appears to generally perform better on the downstream classification metric, and these gains are attributed to better rationale extraction. I think overall the paper represents a valuable contribution but I have some concerns as I lay out below. I will be happy to update my score post rebuttal.

**Limitations And Societal Impact:**

I believe that the broader impact section could use some more work, especially as it relates to the limitations of current work.

**Main Review:**

- I think overall this paper is well motivated and presents an interesting framework that could be extended to various problems (since it does not require human annotations).
- I particularly like the choice of datasets here, along with the extensive analysis as it relates to the error rate associated with the rationale extractor and how it may impact the mutual information between a causal feature and the label and the non causal features and the label.
- I think the results are quite convincing on each dataset (more on Tripadvisor than Beer Review), however, I would have liked to see experiments with more than one rationale extraction framework, especially since multiple methods have been proposed since Lei et al. Additionally, since the counterfactuals are generated based on sampling, I would have liked to see multiple runs of the experiments and means and standard deviations reported in the tables rather than just a single number. This would especially help with the Beer Review results since in many cases certain baselines achieve comparable performance to the proposed method. There are some other outstanding questions as well that the authors haven’t answered. Lastly, the authors say they set the rationale percentage to 10% for all datasets, but don’t offer a justification for this number. What happens if it is 20% or 30%? I imagine the percentage tokens that are relevant for a particular dataset would be dependent on the dataset to some extent (see [1]).
- The paper’s presentation needs to be improved, especially as it relates to Section 2. The authors also appear to use causal terms such as “spurious” to mean something different than what they are. For instance, in Lines 68 and 69, they say “We assume some subset of the features, X_1, belong to the target aspect label, Y_1, while other features, X_2, are spurious.” This sentence implies one of two things: (i) either this is a very strong assumption where all features are confounded; or (ii) the authors meant to say non-causal instead of spurious. Currently I am assuming that it is the latter.
- The paper is lacking a comprehensive qualitative analysis. I think the paper would be considerably better if the authors perform one on generated rationales from various methods (including an error analysis on the same).

[1] Jain, Sarthak, Sarah Wiegreffe, Yuval Pinter, and Byron C. Wallace. "Learning to Faithfully Rationalize by Construction." In Proceedings of the 58th Annual Meeting of the Association for Computational Linguistics, pp. 4459-4473. 2020.

Additional comments:
- In the last paragraph of Page 2 you talk about two reasons why a rationale selector may not work well. There is in fact a third reason too, where the rationale selector (in end to end models) may in fact end up performing the classification task and the classification layers end up as dummy layers. For instance, if a model just extracts a period as the rationale for positive sentiment and any non-period token as rationale for negative sentiment.
- For multiple papers, Arxiv versions have been cited even though the papers have been published at peer reviewed conferences/journals. Please update these references.

Other presentation comments:
- Lines 21 and 22 (‘rationale selector’ and ‘classifier’): In Latex, the opening quotes are ` instead of ‘. Same for Lines 276, 279, and possibly others that I might have missed.
- Use \citet when using an intext citation. For instance, in lines 259 and 261, the paper says “reviews collected by [25]” and “rationalization by [5]” respectively. Instead it should read “reviews collected by Wang et al. [25]” and “rationalization by Chang et al. [5]”
- Line 279: “establish our gains” -> “establish that our gains”


---------------------------------------------------------------------------------------------
I thank the authors for taking the time to respond to my review in such detail. I have updated my score to recommend an acceptance but at the same time I would urge the authors to include the additional experiments I have recommended in my review. I think those will make this a very thorough and influential empirical study.

**Time Spent Reviewing:**

9

---

> ### Author Response · Authors · 2021-08-10
> **Response to reviewer Z6bp**
>
> Thank you for the detailed feedback and reviewer's time.
>
> [Comment]: I would have liked to see experiments with more than one rationale extraction framework. \
> [Response:] We appreciate your thoughtful suggestion. We will conduct empirical comparisons for our future work since this will definitely enhance the generality and robustness of our proposed framework. The ultimate goal of this work is to demonstrate that by lowering the mutual information between spurious and desired signals, our model effectively discovers the rationale for this.  As a result, we chose the MMI baselines since they align most closely with our theoretical model. In general, spurious signals are a problem affecting all rationale extraction methods, and our CDA framework should benefit them.
>
> [Comment:] Multiple Runs. \
> [Response:] We agree that comparing methods with multiple runs with averages and standard deviations would be an ideal way to better illustrate the differences between them. Our experiments are however very expensive because training all the models takes weeks when using our GPU clusters. As soon as the review was received, we started re-running. We expect to obtain the results in September.
>
> [Question]: Rationale percentage related questions \
> [Response:] We set the rationale length at 10% due to previous research with MMI rationale extractors on the same datasets Chang et al 2019 [5]. With respect to questions of varying rationale lengths generally, we believe the question entails asking how the method would behave when rationale lengths are less or greater than ground truth rationales. Qualitatively, we observed the following:
> * When rationale length is set to less than 10%, some of the mutual information between the ground truth tokens may be reduced. In the example rationale "tastes terrific", if the extracted rationale was "terrific" instead, we could still create a correct counterfactual by altering only half of the "ground truth" rationale. This is similar to the example in Figure 5 where not all of the appearance text is selected by the rationale selector because the ‘ground truth’ is greater than 10% of the original text. The CF predictor still generated a valid counterfactual by for example flipping “no” to “excellent”.
> * In cases where rationale length could exceed 10%, we might not reduce mutual information between some spurious text and the target label is not reduced. Nevertheless, the counterfactual predictor may replace the "extra" tokens with the same tokens as the original document or uninformative tokens such as stop words. In our experiments, we observed something like this, and it can greatly help our qualitative analysis.
>
> [Comment]:There is in fact a third reason too, where the rationale selector (in end to end models) may in fact end up performing theclassification task and the classification layers end up as dummy layers. For instance, if a model just extracts aperiod as the rationale for positive sentiment and any non-period token as rationale for negative sentiment. \
> [Response] You make a very important and insightful point by discussing degeneration (dummy classifiers) because in other pieces like [5] this undesirable behaviour has been mentioned as a primary motivation. As our method did not seek to address this issue, we did not include it as a motivation. Due to the importance of this issue, we will include it in future versions of the paper in the last paragraph of page 2.
>
>
> [Comments:] The authors also appear to use causal terms such as “spurious” to mean something different than what they are. \
> [Response:] Thank-you for your notes on spurious vs non-causal. In general, we tried to remain within the information theoretic framework and avoided crossing over into a causal framework. Using non-causal in place of spurious makes this more rigorous, so we will make corresponding changes to the text to reflect your suggestion.
>
> [Comments:]  paper’s presentation, Typos and citations \
> [Response:] Thank you for pointing out the typos and latex errors, and we will fix them in the final version if accepted. The arxiv versions will also replace the published versions. Thanks for providing the citation [JainACL2020]. It will be cited on page 8 of the final version in the related work section.
>
> [Comments:] Limitations And Societal Impact \
> [Response:] A discussion of more limitations will be added to the broader impact section. If accepted, the following similar sentences will be included in the final paper:
> * A limitation to consider is your degeneration comment where the selector makes the classification instead of the classifier.
> * There is a situation  where the initial selecto has a high enough error range that the augmented dataset has the undesirable characteristic of showing a lower mutual information between X_1 and Y_1.
> * Our approach is also limited by the amount of computation required. It is necessary to train a rationale model, a counterfactual predictor, and finally another rationale model in order to train a single classifier.

---

> > ### Comment · Reviewer_Z6bp · 2021-08-18
> > **Thank you for the response!**
> >
> > Thank you for taking the time to write a comprehensive author response. I appreciate your commitment to include additional results in the final version of your paper, and I am happy to recommend acceptance for this paper. I will update my review to reflect this as well.

---

> > > ### Author Response · Authors · 2021-08-19
> > > **Thank-you!**
> > >
> > > We really appreciate your time and suggestions. Thanks!

---

### Official Review · Reviewer_SnqY · 2021-07-17

**Rating:** 8
**Confidence:** 4

**Summary:**

This paper addresses an important problem of correcting the spurious rationales that often get learned due to maximizing the mutual information. Instead, it proposes “counterfactual data augmentation” that lowers the mutual information between spurious signals and the document label. The counterfactual factual data augmentation is an unsupervised method. The paper finally shows strong results on two datasets.


**Main Review:**

1. An important problem to solve and the proposed solution is interesting.
2. The trade-off between lowering the MMI and preserving the fidelity of the rationales is nicely dealt with.
3. Did you use Eraser (April 2020) for evaluating rationales? The precision and F1 metrics look the same but it will be important to confirm that. Maybe you could use that benchmark to show results for two datasets.
4. Check typos such as the incorrect orientation of inverted commas (Line 279, 285, etc.)

**Time Spent Reviewing:**

2

---

> ### Author Response · Authors · 2021-08-10
> **Response to reviewer SnqY**
>
> Your detailed review and your time are greatly appreciated.
>
> [Question:] “Did you use Eraser (April 2020) for evaluating rationales?”  \
> [Response:] Thank you so much for pointing it out. We did not evaluate rationales using Eraser because the paper focuses on multi-aspect datasets, while Eraser is one-aspect dataset.  If accepted, we can make our choice of dataset more explicitly.
>
> [Comment:] Check typos the incorrect orientation of inverted commas \
> [Response:] We will definitely read through the paper carefully to fix the incorrect orientation of inverted commas and other typos.

---

> > ### Comment · Reviewer_SnqY · 2021-08-16
> > **Thank you!**
> >
> > Thank you for your response. Please add the details about the choice of the dataset in the next version. I will stick to my (already positive) score.

---

> > > ### Author Response · Authors · 2021-08-19
> > > **Thanks!**
> > >
> > > We will for sure add details about the dataset choices. Thank-you for your time and notes!

---

### Decision · Program_Chairs · 2021-09-27

**Decision:**

Accept (Poster)

**Comment:**

This paper initially received diverging scores, but after the author response, two reviewers updated their reviews and increased their scores. The reviewers are now unanimous in their assessment that the paper tackles an important problem using an interesting and novel approach, and draws a unique connection between explanations and spurious correlations. It also shows good improvements compared to the baseline.

There are some additional experiments and analyses requested by the reviewers that I urge the authors to add: more extraction methods, more careful experimental settings (multiple runs, report mean and std), other datasets besides the two (the fact there are several versions of them isn't enough), and qualitative analyses.

Finally, while not mentioned by the reviewers, one should acknowledge that the evaluation only targets plausibility (by comparing with human explanations) and not faithfulness. This is especially concerning considering the issues with select-predict models raised by Jacovi and Goldberg, "Aligning Faithful Interpretations with their Social Attribution". Therefore, the paper should at the very least discuss these limitations, and more ideally, also evaluate faithfulness (e.g., via sufficiency and comprehensiveness metrics; see Wiegreffe and Marasović, "Teach Me to Explain: A Review of Datasets for Explainable NLP", for a good summary of these concepts).

Despite these issues, the paper makes valuable contributions and I therefore recommend acceptance.